# WASSERSTEIN FAIR AUTOENCODERS

## ABSTRACT

Autoencoders, or nonlinear factor models parameterized by neural networks, have become an indispensable tool for generative modeling and representation learning in high dimensions. Imposing structural constraints such as conditional independence on the latent variables (representation, or factors) in order to capture invariance or fairness with autoencoders has been attempted through adding *ad hoc* penalties to the loss function mostly in the variational autoencoder (VAE) context, often based on heuristic arguments. In this paper, we demonstrate that Wasserstein autoencoders (WAEs) are highly flexible in embracing structural constraints. Well-known extensions of VAEs for this purpose are gracefully handled within the framework of the seminal result by Tolstikhin et al. (2018). In particular, given a conditional independence structure of the generative model (decoder), corresponding encoder structure and penalties are *induced* from the functional constraints that define the WAE. This property of WAEs opens up a principled way of penalizing autoencoders to impose structural constraints. Utilizing this generative model structure, we present results on fair representation and conditional generation tasks, and compare them with other preceding methods.

## 1 INTRODUCTION

The ability to learn informative representation of data with minimal supervision is a key challenge in machine learning (Tschannen et al., 2018), toward obtaining which autoencoders have become an indispensable toolkit. An autoencoder consists of the encoder, which maps the input to a low-dimensional representation, and the decoder, that maps a representation back to a reconstruction of the input. Thus an autoencoder can be considered a nonlinear factor analysis model as the latent variable provided by the encoder carries the meaning of "representation" and the decoder can be used for generative modeling of the input data distribution. Most autoencoders can be formulated as minimizing some "distance" between the distribution $P_X$ of input random variable $X$ and the distribution $g_\sharp P_Z$ of the reconstruction $G = g(Z)$, where $Z$ is the latent variable or representation having distribution $P_Z$ and $g$ is either deterministic or probabilistic decoder (in the latter case $g$ is read as the conditional distribution of $G$ given $Z$), which is variationally described in terms of an encoder $Q_{Z|X}$. For instance, the variational autoencoder (VAE, Kingma & Welling, 2014) minimizes

$$D_{\text{VAE}}(P_X, g_\sharp P_Z) = \inf_{Q_{Z|X} \in \mathcal{Q}} \mathbb{E}_{P_X} \left[ D_{KL}(Q_{Z|X} \| P_Z) - \mathbb{E}_{Q_{Z|X}} \log g(Z) \right] \tag{1}$$

over the set of probabilistic decoders or conditional densities $g$ of $G$ given $Z$, where $D_{KL}$ is the Kullback-Leibler (KL) divergence, and the Wasserstein autoencoder (WAE, Tolstikhin et al., 2018) minimizes

$$D_{\text{WAE}}(P_X, g_\sharp P_Z) = \inf_{Q_{Z|X} \in \mathcal{Q}} \mathbb{E}_{P_X} \mathbb{E}_{Q_{Z|X}} d^p(X, g(Z)) \tag{2}$$

over the set of deterministic decoders $g$, where $d$ is the metric in the space of input $X$ and $p \geq 1$. Set $\mathcal{Q}$ restricts the search space for the encoder. In VAEs, a popular choice is a class of normal distributions

$$\mathcal{Q} = \{Q_{Z|X} \text{ regular conditional distribution} : Z|\{X = x\} \sim N(\mu(x), \Sigma(x)), \ (\mu, \Sigma) \in \mathcal{NN}\}$$

where $\mathcal{NN}$ is a class of functions parametrized by neural networks. In WAEs, the choice

$$\mathcal{Q} = \{Q_{Z|X} \text{ regular conditional distribution} : Q_Z \triangleq \mathbb{E}_{P_X} Q_{Z|X} = P_Z\} \tag{3}$$

makes the left-hand side of Eq. (2) equal to the ($p$-th power of) the $p$-Wasserstein distance between $P_X$ and $g_\sharp P_Z$ (Tolstikhin et al., 2018, Theorem 1); $Q_Z$ is called an *aggregate posterior* of $Z$. If $\mathcal{Q}$ is

a set of Dirac measures, i.e., $\mathcal{Q} = \{Q_{Z|X} : Q_{Z|X=x} = \delta_{f(x)}, \; f \in \mathcal{NN}\}$, then minimizing Eq. (2) reduces to the learning problem of a deterministic unregularized autoencoder.

Of course, the notion of "informativeness" depends on the downstream task. The variation in the observations that are not relevant to the particular task is often called "nuisance" and is desirable to be suppressed from the representation. For example, in finding "fair representations," (Zemel, 2013) sensitive information such as gender or socioeconomic status should be removed from latent representations; in obtaining representations of facial images, those that are invariant to lighting conditions, poses, or wearing of eyeglasses are often sought. A popular approach to this goal is to explicitly separate informative and nuisance variables in the generative model by factorization. This approach imposes a structure on the decoder. Additionally the encoder is further factorized and a penalty promoting independence between the encoded representation and nuisance variable can be added. A well-known example is the variational fair autoencoder (VFAE, Louizos et al., 2016), in which a variant of the "M1+M2" graphical model (Kingma et al., 2014) is used to factorize the decoder and a resembling factorization of the encoder (variational posterior) is assumed. Independence of the representation from nuisance variable is encouraged by adding a maximum mean discrepancy (MMD, Gretton et al., 2007) between conditional variational posteriors; in Lopez et al. (2018), MMD is replaced by the Hilbert-Schmidt Independence Criterion (HSIC, Gretton et al., 2007). Other authors employ penalties derived from the mutual information (MI) (Moyer et al., 2018; Song et al., 2019; Creager et al., 2019). Another example is the Fader Networks (Lample et al., 2018), in which the deterministic decoder takes an additional input of the attribute (such as whether or not eyeglasses are present in a portrait) and an adversarial penalty that hinders the accurate prediction of the attribute by the deterministic, unfactorized encoder.

These examples illustrate that, while the generative model (decoder structure) can be chosen suitably for the downstream task, there is no principled way of imposing the corresponding encoder structure. In this paper, we show that the WAE framework allows us to *automatically determine* the encoder structure corresponding to imposed decoder structure. Specifically, when the deterministic decoder $g$ in Eq. (2) is modified to handle the conditional independence structure of the imposed generative model, then the constraint set (amounting to the $\mathcal{Q}$ in Eq. (3)) that makes the LHS of Eq. (2) a proper (power of) Wasserstein distance determines the factorization of the (deterministic) encoder. In practice, the hard constraints in $\mathcal{Q}$ is relaxed and Eq. (2) is solved in a penalized form. Following the approach of Tolstikhin et al. (2018), the cited constraint set can be systemically translated to penalties. Therefore, in addition to the theoretical advantage that the penalized form equals a genuine distributional distance for sufficiently large penalty parameter while that of Eq. (1) remains a lower bound of the negative log-likelihood of the model, the *ad hoc* manner of designing penalties prevalent in the VAE literature can be avoided in the WAE framework. Further, the allowance of deterministic encoder/decoder promotes better generation performance in many downstream tasks.

We explain how the WAE framework leads to structured encoders given a generative model through examples reflecting downstream tasks in Sect. 3 after providing necessary background in Sect. 2. We would call these structured uses of WAEs the *Wasserstein Fair Autoencoders* (WFAEs). After reviewing relevant ideas in Sect. 4, WFAEs are experimented in Sect. 5 for datasets including VGGFace2 (Cao et al., 2018). We conclude the paper in Sect. 6.

## 2  PRELIMINARIES

In fitting a given probability distribution $P_X$ of a random variable $X$ on a measurable space $(\mathcal{X}, \mathcal{B}(\mathcal{X}))$, where $\mathcal{X} \subset \mathbb{R}^D$ equipped with metric $d$, by a generative model $P_G$ of sample $G$ on the same measurable space, one may consider minimizing the ($p$th power of) $p$-Wasserstein distance between the two distributions, i.e.,

$$\min_{P_G \in \mathcal{M}} \left\{ W_p^p(P_X, P_G) := \inf_{\pi \in \mathcal{P}(P_X, P_G)} \mathbb{E}_\pi \, d^p(X, G) \right\}.$$

Here, $\mathcal{M}$ is the model space of probability distributions, $\mathcal{P}(P_X, P_G)$ is the *coupling* or the set of all joint distributions on $(\mathcal{X} \times \mathcal{X}, \mathcal{B}(\mathcal{X} \times \mathcal{X}))$ having marginals $P_X$ and $P_G$. Often the sample $G$ is generated by transforming a variable in a latent space. When $G = g(Z)$ a.s. for a latent variable $Z$ in a probability space $(\mathcal{Z}, \mathcal{B}(\mathcal{Z}), P_Z)$, $\mathcal{Z} \subset \mathbb{R}^l$, and measurable function $g$, then $P_G$ is denoted by $g_\sharp P_Z$, where $\sharp$ is the *push forward* operator. In this setting, as discussed in Sect. 1, Tolstikhin et al.

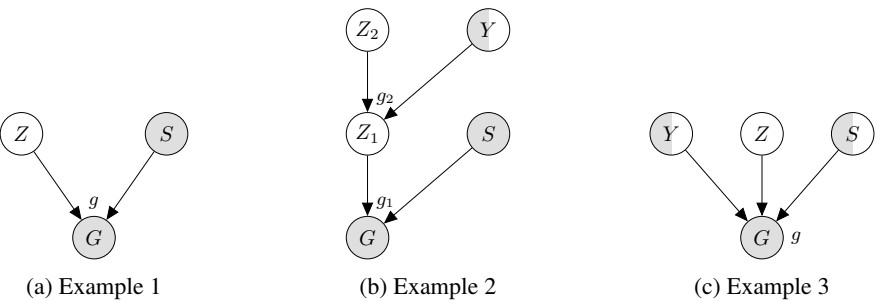

Figure 1: Examples of generative models for WFAEs

(2018) show that $W_p^p(P_X, g_\sharp P_Z) = D_{\text{WAE}}(P_X, g_\sharp P_Z)$ (Eq. (2)), with the constraint set $\mathcal{Q}$ on the probabilistic encoders $Q_{Z|X}$ given in Eq. (3). It is further claimed by Patrini et al. (2020) that the set of conditional distributions $Q_{Z|X}$ can be reduced to be deterministic, i.e., $Z = f(X)$ a.s. for $f$ measurable. However, this claim is not in general true unless $g$ is *injective*:

**Theorem 1** *Let $d(x, y) = \|x - y\|_2$ for $x, y \in \mathcal{X}$. If $P_X$ has a density with respect to the Lebesgue measure, and the measurable function $g : \mathcal{Z} \to \mathcal{X}$ is injective, then*

$$W_2^2(P_X, g_\sharp P_Z) = \inf_{f \in \mathcal{Q}} \mathbb{E}_{P_X} d^2(X, g(f(X))),\tag{4}$$

*where $\mathcal{Q}$ is the set of all measurable functions from $\mathcal{X}$ to $\mathcal{Z}$ such that $f_\sharp P_X = P_Z$.*

The proof of this result is provided in Appendix A of the Supplement.

**Remark 1** *In Patrini et al. (2020, Theorem A.2), it is incorrectly claimed that for the right inverse $\tilde{g}$ of $g$ when the codomain of the latter is restricted to its range, $(\tilde{g} \circ g)_\sharp P_Z(F)$ is equal to $P_Z(\tilde{g}^{-1}(g^{-1}(F)))$, instead of the correct $P_Z(g^{-1}(\tilde{g}^{-1}(F)))$. This confusion invalidates the rest of the argument of the cited theorem.*

In practice the set $\mathcal{Q}$ can be relaxed to $\mathcal{F}$, a class of all measurable functions parameterized by deep neural networks, which contains a minimizer of the right-hand side (RHS) of Eq. (4); the constraint $f_\sharp P_X = P_Z$ can be met by adding a penalty $\lambda \mathcal{D}(f_\sharp P_X \| P_Z)$ for sufficiently large multiplier $\lambda > 0$ and a divergence $\mathcal{D}$ between two distribution. Thus if we define the distortion criterion

$$\delta(f, g) = \mathbb{E}_{P_X} d^2(X, g(f(X))) + \lambda \mathcal{D}(f_\sharp P_X \| P_Z),$$

then the generative modeling problem based on 2-Wasserstein distance can be formulated as

$$\inf_{g \in \mathcal{G}} \inf_{f \in \mathcal{F}} \delta(f, g),\tag{5}$$

for $\mathcal{G}$ a set of injective measurable functions from $\mathcal{Z}$ to $\mathcal{X}$, typically parameterized by deep neural networks. The function $f : \mathcal{X} \to \mathcal{Z}$ has an interpretation of an *encoder* and $g : \mathcal{Z} \to \mathcal{X}$ has an interpretation of a *decoder*. Typically $l \ll D$.

## 3 LEARNING INVARIANT REPRESENTATIONS WITH WFAES

Often generative modeling is more complicated than just involving a latent variable $Z$ in $\mathcal{Z}$ and its reconstruction $G$ in $\mathcal{X}$. For example, data may come with labels, which can be employed in the generation process to learn *invariant representations*.

**Example 1** *Let us begin with a simple generative model shown in Fig. 1a (Louizos et al., 2016, Fig. 1); see also (Kingma et al., 2014, M2). Here, variable $S \in \mathcal{S} \subset \mathbb{R}^B$ represents the observed nuisance variation, and $Z$ models the remaining information on $G$ (with which we want to mimic the observable variable $X$) that is independent of $S$. Thus the $Z$ encodes the representation* invariant *to the unwanted variation in $S$. Denoting the marginal distribution of the nuisance variable $S$ by $P_S$, the distribution of model $G$ is $g_\sharp(P_Z \otimes P_S)$, where $\otimes$ is used to denote a product distribution. The goal is to make the joint distribution $P_{GS}$ of $(G, S)$ close to $P_{XS}$ of $(X, S)$, the observable.*

If we let $\tilde{g}(z, s) = (g(z, s), s)$, then $P_{GS} = \tilde{g}_\sharp(P_Z \otimes P_S)$. Recall that $\mathcal{X}$ is equipped with metric $d$. Equip $\mathcal{S}$ with another metric $d'$ and $\mathcal{X} \times \mathcal{S}$ with $\tilde{d} = \sqrt{d^2 + (d')^2}$. Then, by applying Theorem 1 to $P_{XS}$, $P_Z \otimes P_S$, and $\tilde{g}$, we obtain

$$W_2^2(P_{XS}, \tilde{g}_\sharp(P_Z \otimes P_S)) = \inf_{\tilde{f} \in \tilde{\mathcal{F}}} \mathbb{E}_{P_{XS}} \tilde{d}^2\left([X, S], \tilde{g}(\tilde{f}(X, S))\right)$$

$$= \inf_{f \in \mathcal{F}} \mathbb{E}_{P_{XS}} \tilde{d}^2\left([X, S], [g(f(X, S), S), S]\right) = \inf_{f \in \mathcal{F}} \mathbb{E}_{P_{XS}} d^2\left(X, g(f(X, S), S)\right),$$

where $\tilde{\mathcal{F}} = \{\tilde{f} : \mathcal{X} \times \mathcal{S} \to \mathcal{Z} \times \mathcal{S} : \tilde{f}_\sharp P_{XS} = P_Z \otimes P_S\}$, $\mathcal{F} = \{f : \mathcal{X} \times \mathcal{S} \to \mathcal{Z} : (f, \Pi_S)_\sharp P_{XS} = P_Z \otimes P_S\}$, and $\Pi_S : \mathcal{X} \times \mathcal{S} \to \mathcal{S} : \Pi_S(x, s) = s$ is the orthogonal projection from $\mathcal{X} \times \mathcal{S}$ onto $\mathcal{S}$. The second equality holds by noting that $\tilde{f}(x, s) = (f(x, s), h(x, s))$ and taking $h = \Pi_S$. The latter constraint set $\mathcal{F}$ means that

$$f(X, S) \stackrel{d}{=} Z, \quad f(X, S) \perp\!\!\!\perp S. \tag{6}$$

Following formulation equation 5 for the unstructured case, we can incorporate constraint equation 6 into the learning problem in a penalized form

$$\min_g \min_f \mathbb{E}_{P_{XS}} d^2\left(X, g(f(X, S), S)\right) + \lambda_1 \mathcal{D}(f_\sharp P_{XS} \| P_Z) + \lambda_2 \mathcal{H}((f, \Pi_S)_\sharp P_{XS}),$$

where $\mathcal{D}$ is an appropriate divergence between two probability distributions such as MMD or the generative adversarial network (GAN) loss as suggested by Tolstikhin et al. (2018), and $\mathcal{H}$ promotes independence between two random variables $f(X, S)$ and $S$, such as the HSIC (Lopez et al., 2018).

**Example 2** *Consider a more involved generative model shown in Fig. 1b, which is employed by the VFAE (Louizos et al., 2016, Fig. 2) as an extension of the "M1 + M2" semi-supervised model (Kingma et al., 2014). This graphical model actually describes the* conditional *distribution $P_{X|S}$ of $X$ given $S$, since $S$ and $Y$ are allowed to be correlated. Instead, it is required*

$$Z_1 \perp\!\!\!\perp S \tag{7}$$

*in order to impose invariance to the nuisance variable $S$. Let $g : \mathcal{Y} \times \mathcal{Z}_2 \times \mathcal{S} \to \mathcal{X} \times \mathcal{Y} \times \mathcal{S}$ as $g(y, z_2, s) = \left(g_1(g_2(y, z_2), s), y, s\right)$. Denoting the marginal distribution of the nuisance variable $S$ by $P_S$ and the joint distribution of $Y$ and $S$ by $P_{YS}$, the distribution of model $G$ is $g_\sharp(P_{YS} \otimes P_{Z_2})$. The goal is to make the joint distribution $P_{GS}$ of $(G, S)$ close to $P_{XS}$ of $(X, S)$ when $Y$ is not observed, and $P_{GYS}$ of $(G, Y, S)$ to $P_{XYS}$ of $(X, Y, S)$ when the data is fully observed.*

First consider the case that $Y$ is missing. Let $\Pi_{XS}$ be the orthogonal projection operator from $\mathcal{X} \times \mathcal{Y} \times \mathcal{S}$ onto $\mathcal{X} \times \mathcal{S}$. Then by applying Theorem 1 to $P_{XS}$, $P_{YS} \otimes P_{Z_2}$, and $\Pi_{XS}g$, we obtain

$$W_2^2(P_{XS}, \Pi_{XS}g_\sharp(P_{YS} \otimes P_{Z_2})) = \inf_{f \in \mathcal{F}_{\text{unobs}}} \mathbb{E}_{P_{XS}} \tilde{d}^2\left([X, S], \Pi_{XS}g(f(X, S), S)\right)$$

$$= \inf_{f \in \mathcal{F}_{\text{unobs}}} \mathbb{E}_{P_{XS}} \tilde{d}^2\left([X, S], [g_1(g_2(f(X, S)), S), S]\right) = \inf_{f \in \mathcal{F}_{\text{unobs}}} \mathbb{E}_{P_{XS}} d^2(X, g_1(g_2(f(X, S)), S)),$$

where $\mathcal{F}_{\text{unobs}} = \{(f_1^{\text{unobs}}, f_2^{\text{unobs}}) | f_1^{\text{unobs}} : \mathcal{X} \times \mathcal{S} \to \mathcal{Y}, f_2^{\text{unobs}} : \mathcal{X} \times \mathcal{S} \to \mathcal{Z}_2, (f_1^{\text{unobs}}, \Pi_S, f_2^{\text{unobs}})_\sharp P_{XS} = P_{YS} \otimes P_{Z_2}\}$. The latter constraint set means

$$(f_1^{\text{unobs}}(X, S), S) \stackrel{d}{=} (Y, S), \quad f_2^{\text{unobs}}(X, S) \stackrel{d}{=} Z_2, \quad (f_1^{\text{unobs}}(X, S), S) \perp\!\!\!\perp f_2^{\text{unobs}}(X, S). \tag{8}$$

Now consider the case $Y$ is observed. Equip $\mathcal{Y}$ with a metric $d''$ and $\mathcal{X} \times \mathcal{Y} \times \mathcal{S}$ with $\breve{d} = \sqrt{d^2 + (d'')^2 + (d')^2}$. Then by applying Theorem 1 to $P_{XYS}$, $P_{YS} \otimes P_{Z_2}$, and $g$, we obtain

$$W_2^2(P_{XYS}, g_\sharp(P_{YS} \otimes P_{Z_2})) = \inf_{f_2^{\text{obs}} \in \mathcal{F}_{\text{obs}}} \mathbb{E}_{P_{XYS}} \breve{d}^2\left([X, Y, S], g(Y, f_2^{\text{obs}}(X, Y, S), S)\right)$$

$$= \inf_{f_2^{\text{obs}} \in \mathcal{F}_{\text{obs}}} \mathbb{E}_{P_{XYS}} \breve{d}^2\left([X, Y, S], [g_1(g_2(Y, f_2^{\text{obs}}(X, Y, S)), S), Y, S]\right)$$

$$= \inf_{f_2^{\text{obs}} \in \mathcal{F}_{\text{obs}}} \mathbb{E}_{P_{XYS}} d^2(X, g_1(g_2(Y, f_2^{\text{obs}}(X, Y, S)), S)),$$

where $\mathcal{F}_{\text{obs}} = \{f_2^{\text{obs}} : \mathcal{X} \times \mathcal{Y} \times \mathcal{S} \to \mathcal{Z}_2 : (\Pi_Y, \Pi_S, f_2^{\text{obs}})_\sharp P_{XYS} = P_{YS} \otimes P_{Z_2}\}$ and $\Pi_Y : (x, y, s) \mapsto y$, $\Pi_S : (x, y, s) \mapsto s$ are projections. The latter constraint set means

$$f_2^{\text{obs}}(X, Y, S) \stackrel{d}{=} Z_2, \quad (Y, S) \perp\!\!\!\perp f_2^{\text{obs}}(X, Y, S). \tag{9}$$

In order to combine the two Wasserstein losses and constraints Eq. (7) to (9), let us extend $\mathcal{Y}$ to $\bar{\mathcal{Y}} = \mathcal{Y} \cup \{*\}$, where '$*$' represents the missing value. For any $(f_1^{\text{unobs}}, f_2^{\text{unobs}}) \in \mathcal{F}_{\text{unobs}}$ and $f_2^{\text{obs}} \in \mathcal{F}_{\text{obs}}$, define $f_1 : \mathcal{X} \times \bar{\mathcal{Y}} \times \mathcal{S} \to \mathcal{Y}$ and $f_2 : \mathcal{X} \times \bar{\mathcal{Y}} \times \mathcal{S} \to \mathcal{Z}_2$ as

$$f_1(x, y, s) = \begin{cases} y, & y \neq *, \\ f_1^{\text{unobs}}(x, s), & y = *, \end{cases} \qquad f_2(x, y, s) = \begin{cases} f_2^{\text{obs}}(x, y, s), & y \neq *, \\ f_2^{\text{unobs}}(x, s), & y = *. \end{cases}$$

Then we can formulate the learning problem for the WFAE in a penalized form

$$\min_{g_1, g_2} \min_{f_1, f_2} \mathbb{E}_{P_{XYS}} d^2\Big(X, g_1\big(g_2(Y, f_2(X, Y, S)), S\big)\Big) + \mathbb{E}_{P_{XS}} d^2\Big(X, g_1\big(g_2(f_1(X, *, S), f_2(X, *, S)), S\big)\Big)$$
$$+ \lambda_1 \mathcal{D}_1((f_1, \Pi_S)_\sharp P_{XYS} \| P_{YS}) + \lambda_2 \mathcal{D}_2(f_{2\sharp} P_{XYS} \| P_{Z_2}) + \lambda_3 \mathcal{H}_3((f_1, \Pi_S, f_2)_\sharp P_{XYS})$$
$$+ \lambda_4 \mathcal{H}_4((g_2 \star f_1)_\sharp (P_{XYS} \otimes P_{Z_2})),$$

where $g_2 \star f_1(x, y, z_2, s) = (g_2(f_1(x, y, s), z_2), s)$; $\mathcal{D}_1$ and $\mathcal{D}_2$ are appropriate divergences between two probability distributions, and $\mathcal{H}_3$, $\mathcal{H}_4$ promotes independence between two random variables. Note, unlike Example 1 in which only the encoder $f$ is constrained, Eq. (7) imposes a constraint on the decoder $g_2$. Also note that, the divergence $\mathcal{D}_1$ can be estimated in a two-sample fashion, namely from the a sample drawn from $P_{YS}$, i.e., $(y_i, s_i)$ with $y_i$ observed, and another sample drawn from $(f_1, \Pi_S)_\sharp P_{XYS}$, either as $(y_j, s_j)$ if $y_j$ is observed or $(f_1(x_j, *, s_j), s_j)$ otherwise. Hence all the data from the minibatch can be utilized. Likewise, divergence $\mathcal{D}_2$ and the independence penalties $\mathcal{H}_3$ and $\mathcal{H}_4$ can utilize the full minibatch.

**Remark 2** *VAE-based models, e.g., VFAE, assume a specific factorization of the variational posterior (encoder). Since the factor $q_\phi(y|z)$ for imputing $Y$ does not appear in the evidence lower bound (ELBO) of the observed likelihood, an additional penalty on this factor evaluated for the fully observed sample is coined (Louizos et al., 2016, Eq. 5), making the bound not tight. In the WFAE, on the contrary, the $\mathcal{D}_1$ term that arises naturally from constraint equation 8 for the Wasserstein distance penalizes the imputing encoder $f_1$ for both fully (by requiring $f_1(x_j, y_j, s_j) = y_j$) and partially (by the divergence) observed samples.*

**Example 3** *The model shown in Fig. 1c extends Example 1 with two independent nuisance variables that can be missing. Here $Y$ may represent a person's identity in her portrait, which may be missing, and $S$ partially observed attributes (e.g., sunglasses on/off, mouth open/closed, and gender). In this setup we want two different portraits of a person to have similar values of $Z$, and those of two different people to have quite distinct values of $Z$, even if the encoder does not know whose portraits they are. We may also want $Z$ to represent something immune even to gender switch.*

Proceeding as Example 2, we obtain for $\tilde{g} : \mathcal{Y} \times \mathcal{Z} \times \mathcal{S} \to \mathcal{X} \times \mathcal{Y} \times \mathcal{S}$: $\tilde{g}(y, z, s) = (g(y, z, s), y, s)$,

$$W_2^2(P_X, \Pi_X \tilde{g}_\sharp (P_Y \otimes P_Z \otimes P_S)) = \inf_{(f_1^X, f_2^X, f_3^X) \in \mathcal{F}^X} \mathbb{E}_{P_X} d^2(X, g(f_1^X(X), f_2^X(X), f_3^X(X))),$$

$$\mathcal{F}^X = \{(f_1, f_2, f_3) : (f_1, f_2, f_3)_\sharp P_X = P_Y \otimes P_Z \otimes P_S, \ f_1 : \mathcal{X} \to \mathcal{Y}, \ f_2 : \mathcal{X} \to \mathcal{Z}, \ f_3 : \mathcal{X} \to \mathcal{S}\}$$

when both $Y$ and $S$ are unobserved,

$$W_2^2(P_{XS}, \Pi_{XS} \tilde{g}_\sharp (P_Y \otimes P_Z \otimes P_S)) = \inf_{(f_1^{XS}, f_2^{XS}) \in \mathcal{F}^{XS}} \mathbb{E}_{P_{XS}} d^2(X, g(f_1^{XS}(X, S), f_2^{XS}(X, S), S)),$$

$$\mathcal{F}^{XS} = \{(f_1, f_2) : (f_1, f_2, \Pi_S)_\sharp P_{XS} = P_Y \otimes P_Z \otimes P_S, \ f_1 : \mathcal{X} \times \mathcal{S} \to \mathcal{Y}, \ f_2 : \mathcal{X} \times \mathcal{S} \to \mathcal{Z}\}$$

when only $Y$ is unobserved,

$$W_2^2(P_{XY}, \Pi_{XY} \tilde{g}_\sharp (P_Y \otimes P_Z \otimes P_S)) = \inf_{(f_2^{XY}, f_3^{XY}) \in \mathcal{F}^{XY}} \mathbb{E}_{P_{XY}} d^2(X, g(Y, f_2^{XY}(X, Y), f_3^{XY}(X, Y))),$$

$$\mathcal{F}^{XY} = \{(f_2, f_3) : (\Pi_Y, f_2, f_3)_\sharp P_{XY} = P_Y \otimes P_Z \otimes P_S, \ f_2 : \mathcal{X} \times \mathcal{Y} \to \mathcal{Z}, \ f_3 : \mathcal{X} \times \mathcal{Y} \to \mathcal{S}\}$$

when only $S$ is unobserved, and

$$W_2^2(P_{XYS}, \tilde{g}_\sharp(P_Y \otimes P_Z \otimes P_S)) = \inf_{f_3^{XYS} \in \mathcal{F}^{XYS}} \mathbb{E}_{P_{XYS}} d^2(X, g(Y, f_2^{XYS}(X, Y, S), S)),$$

$$\mathcal{F}^{XYS} = \{f_2 : (\Pi_Y, f_2, \Pi_S)_\sharp P_{XYS} = P_Y \otimes P_Z \otimes P_S, \ f_2 : \mathcal{X} \times \mathcal{Z} \to \mathcal{S}\}$$

when the data are fully observed. If we expand $\mathcal{Y}$ to $\bar{\mathcal{Y}} = \mathcal{Y} \cup \{*\}$ and $\mathcal{S}$ to $\bar{\mathcal{S}} = \mathcal{S} \cup \{*\}$, then the learning problem is

$$\min_g \min_{(f_1, f_2, f_3) \in \mathcal{F}} \mathbb{E}_{P_{XYS}} d^2\big(X, g\big(Y, f_2(X, Y, S), S\big)\big) + \mathbb{E}_{P_{XY}} d^2\big(X, g\big(Y, f_2(X, Y, *), f_3(X, Y, *)\big)\big)$$

$$+ \mathbb{E}_{P_{XS}} d^2\big(X, g\big(f_1(X, *, S), f_2(X, *, S), S\big)\big)$$

$$+ \mathbb{E}_{P_X} d^2\big(X, g\big(f_1(X, *, *), f_2(X, *, *), f_3(X, *, *)\big)\big)$$

$$+ \lambda_1 \mathcal{D}_1(f_{1\sharp} P_{XYS} \| P_Y) + \lambda_2 \mathcal{D}_2(f_{2\sharp} P_{XYS} \| P_Z) + \lambda_3 \mathcal{D}_3(f_{3\sharp} P_{XYS} \| P_S)$$

$$+ \lambda_4 \mathcal{H}_4((f_1, f_2, f_3)_\sharp P_{XYS}),$$

where $\mathcal{H}_4$ measures dependence of three random variables, e.g., the $d$-variate HSIC (Lopez et al., 2018) with $d = 3$, and $\mathcal{F} = \{(f_1, f_2, f_3) : f_1 : \mathcal{X} \times \bar{\mathcal{Y}} \times \bar{\mathcal{S}} \to \mathcal{Y}, f_2 : \mathcal{X} \times \bar{\mathcal{Y}} \times \bar{\mathcal{S}} \to \mathcal{Z}, f_3 : \mathcal{X} \times \bar{\mathcal{Y}} \times \bar{\mathcal{S}} \to \mathcal{S}\}$, for

$$f_1(x, y, s) = \begin{cases} f_1^X(x), & y = *, s = *, \\ f_1^{XS}(x, s), & y = *, s \neq *, \\ y, & y \neq *, \end{cases} \quad f_3(x, y, s) = \begin{cases} f_3^X(x), & y = *, s = *, \\ f_3^{XY}(x, y), & y \neq *, s = *, \\ s, & s \neq *, \end{cases}$$

and $f_2(x, y, s)$ is equal to $f_2^X(x)$ if $y = *, s = *$, to $f_2^{XY}(x, y)$ if $y \neq *, s = *$, to $f_2^{XS}(x, s)$ if $y = *, s \neq *$, and to $f_2^{XYS}(x, y, s)$ otherwise.

**Remark 3** *If variable $Y$ is removed and $S$ is fully observed, Example 3 reduces to Example 1, where the $f_2(x, y, s)$ from the former corresponds to the $f(x, s)$ from the latter. The Fader Networks (Lample et al., 2018) implicitly use this model to obtain attribute-invariant representations of facial images. The adversarial penalty for training the network (Lample et al., 2018, Eq. 2) can be understood as promoting independence between $S$ and $\hat{Z} = f(X, S)$. While in Lample et al. (2018) the encoder $f$ does not depend on $S$, Example 1 shows that it is more natural to take $S$ as an input to remove its effect on $\hat{Z}$. Example 3 can be considered a generalization of the Fader Networks for missing attributes and unknown identities.*

## 4 RELATED WORK

The literature on VAEs is vast. $\beta$-VAE (Higgins et al., 2017) is one of well-known ways of adding penalties to the ELBO of a VAE, which adds one proportional to the expected KL divergence between the variational posterior (encoder) and prior $P_Z$. It is observed that this penalty promotes factorization of the aggregate posterior $Q_Z$ (Kim & Mnih, 2018). In fair representation, VFAE (Louizos et al., 2016) and HSIC-constrained VAE (HCV, Lopez et al., 2018) add penalties to the ELBO for semi-supervised disentanglement along this line. Adversarial penalties have been also considered (Edwards & Storkey, 2016; Madras et al., 2018). Song et al. (2019) bring an information-theoretic interpretation to these approaches. In this regard, penalizing MI between nuisance variable $S$ and encoded latent variable $Z$ (Moyer et al., 2018; Song et al., 2019; Creager et al., 2019) or its tractable upper bounds, e.g., based on a variational approximation (Rodríguez-Gálvez et al., 2021), has been advocated. Recently proposed FairDisCo (Liu et al., 2022) uses the $L_2$ distance between the joint density of $S$ and $Z$ and the product density of their marginals, showing its asymptotic equivalence to the MI. However, as stated in Sect. 1, these penalties promoting desired structures are chosen rather *ad hoc* and loosens the already-not-tight ELBO. Furthermore, there is no principle for choosing the encoder structure corresponding the imposed decoder structure.

The WAE framework discussed in the previous section can overcome these pitfalls in VAEs. The WAE literature has focused on improving the divergence in the penalized form of Eq. (2) that matches the prior $P_Z$ and the aggregated posterior. The original proposal by Tolstikhin et al. (2018) is to employ either the MMD or GAN divergence. Kolouri et al. (2019) propose to use the sliced Wasserstein distance in order to simplify computation. Patrini et al. (2020) consider the Sinkhorn

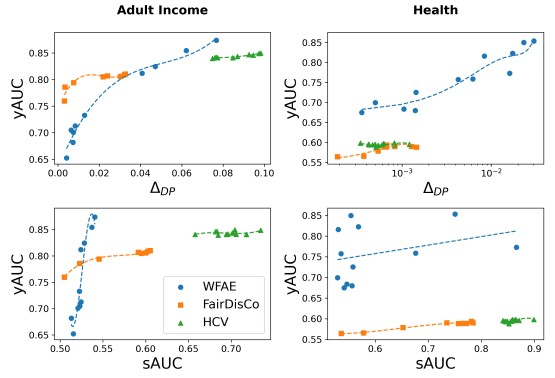

Figure 2: Fair representation trade-off.

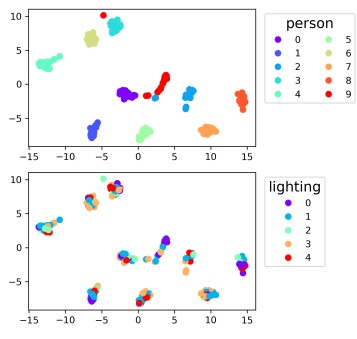

Figure 3: T-SNE map of $Z_1$ in Extended Yale B

divergence (Genevay et al., 2018), computing of which can be boosted by using the Sinkhorn algorithm (Cuturi, 2013). Xu et al. (2020) and Nguyen et al. (2021) propose and improve the relational divergence called the fused Gromov-Wasserstein distance. The latter three works consider the setting in which the prior $P_Z$ is structured. In contrast, we focus on the setting in which the decoder is structured and nuisance information is (partially) available. According to the taxonomy of Tschannen et al. (2018), the former is close to the clustering meta-prior whereas the latter is close to the disentangling one. We emphasize that the cited divergences are compatible with our framework.

## 5 EXPERIMENTS

We experimented WFAEs with various real-world datasets. The generative models for these datasets mainly follow Examples 2 and 3, in most of which variable $Y$ (and sometimes $S$) has the meaning of a label and thus categorical. In order to embed this variable to the Euclidean space $\mathbb{R}^B$ where $B$ does not necessarily depend on the number of categories, we employed the entity embedding network (Guo & Berkhahn, 2016) for observed labels. The trained embedding network naturally becomes a pretrained encoder $f_1$ or $f_3$ from Examples 2 and 3. A by-product of this embedding is that it is even possible to impute categories not present in training data.

### 5.1 FAIR REPRESENTATIONS

To demonstrate the performance of WFAEs on fair representation, we reproduced experiments in Liu et al. (2022) using two categorical datasets, namely the Adult Income and Health datasets. Refer to the appendix for data summary and network implementation. The generative model for the WFAE was the structure of Example 2. With the $Z_1 = g_2(Y, f_2(X, Y, S))$ encoded from the trained model, we quantified the trade-off between fairness and utility (Zhao et al., 2017): we classified $S$ and $Y$ using random forest method, calculated the area under the ROC curve (AUC) on the test data (sAUC and yAUC) as a function of demographic parity $\Delta_{DP}$, and compared the performance with the HSIC-constrained VFAE (HCV) and the FairDisCo. The results are summarized in Fig. 2. While WFAE shows a clear trade-off, other methods are relatively insensitive to demographic parity.

### 5.2 INVARIANT REPRESENTATIONS

The same structure as Sect. 5.1 is used to test the ability of WFAEs to learn invariant representations of controlled photographs. The cropped Extended Yale B dataset Georghiades et al. (2001); Lee et al. (2005) comprises of facial images of 38 human subjects in various lighting conditions. For each subject, the pictures of the person are split into training and test data with a fixed ratio, resulting in 1,664 and 750 images for the training and test respectively. We set the identity of the image as $Y$ and the lighting condition (elevation and azimuth of the light direction normalized into $[-1, 1] \times [-1, 1]$) as $S$. In the training stage, we first trained $f_1$ to estimate $Y$, then trained the rest of the network with $f_1$ held fixed. In consequence, we were able to encode and decode the test data without the information about $Y$ by replacing it with $f_1(X, *, S)$. Although we trained the model

| Net-work | ID accu-racy | Lighting group (Acc.) | | Lighting direction (MSE) | |
|---|---|---|---|---|---|
| | | RF | Logistic | RF | Linear |
| VAE | 0.71 | 0.74 | 0.74 | 0.03 | 0.07 |
| HCV | 0.75 | 0.60 | 0.45 | 0.13 | 0.22 |
| FairDisCo | **1.0** | **0.34** | 0.37 | - | - |
| WFAE | 0.97 | **0.35** | **0.28** | **0.23** | **0.23** |

Table 1: Invariant representation of Extended Yale B. RF=random forest, Logistic=logistic regression, Linear=linear regression. Classification accuracy for discrete and mean squared error for continuous variables.

| Network | Extended Yale B | | MNIST | |
|---|---|---|---|---|
| | Sharpness | FID | Sharpness | FID |
| FairDisCo | 3.36e-4 | 63.7 | 3.45e-2 | 111.1 |
| WFAE | 4.43e-4 | 66.5 | 9.79e-2 | 19.4 |
| Test Data | 3.09e-3 | - | 1.83e-1 | - |

Table 2: Sample generation quality measures.

with continuous $S$, we present some of the results with S categorized in 5 directions, as in Lopez et al. (2018). The results are presented Table 1. The $Z_1$ encoded by WFAE shows better performance in predicting $Y$ and worse in predicting $S$ than others, suggesting better invariant representation. The t-SNE visualization of $Z_1$ in Fig. 3 accords with this result, showing noticeable separation with respect to $Y$, but not with respect to $S$. In panel A of Fig. 4 (top left), the green box depicts generated images by encoding the test image $X$ and nuisance data $S$ into $Z_1 = g_2(f_1(X, *, S), f_2(X, *, S))$, and then computing $g_1(Z_1, S)$. Those in the red box were generated by using the same $Z_1$ but setting $S = (\pm 0.3, \pm 0.3)$. WFAE produced reconstructions closer to the input than HCV and FairDisCo, and perturbing $S$ only kept the identity of the input in the generated images. The sharpness and the Fréchet inception distance (FID) scores are shown Table 2 to assess the sample generation quality. WFAE produced sharper images than FairDisCo, confirming the visual inspection. The FID scores should be taken with caution, though. Since the sample generation is conducted by varying the "lighting direction" attribute (considered as the $S$ variable) the generated samples should be different from the test data with scarce images. Rather, it may indicate samples generated from FairDisCo is less sensitive to $S$, which can also be verified visually.

## 5.3 CONDITIONAL GENERATION

We further investigated the conditional generation capability of WFAEs using the MNIST and VG-GFace2 datasets (Cao et al., 2018).

**MNIST**. We treated the digit attribute as $S$. The generative model for the data is similar to Example 3, but without $Y$. We first trained encoder $f_3$ that estimates $S$, then trained the rest of the network. The final network was tested with images with digit information removed. We also trained a network without the encoder $f_2$ for $Z$, hence it decodes an image using only estimated $S$. Fig. 4 summarize the results, all of which were generated from test data without information of $S$. Penal A (top right) shows decoded samples from $g$ with estimated $S$ and i) not using $f_2$ (blue box), ii) using encoded $Z = f_2(X, S)$ from the test data (green box), and iii) using $Z$ sampled from prior $P_Z$ (red box). Decoded images with the same $S$ all retained their digit information. Reconstruction without using $f_3$, although recognizable, produced degraded images, implying loss of information. FairDisCo with a similar architecture produced quite degraded results; see also Table 2.[1] In panel B (top), we estimated $Z$ from the source and $S$ from the target and generated new images by $g(Z, S)$.

**VGGFace2**. This dataset contains 3.14M training images of faces of total 8631 subjects and 169k test images of total 500 subjects, with partially observed binary attributes such as gender, wearing of sunglasses, and openness of mouth, available for a subset of 30,000 images. Here, we treat the identity of the image as class $Y$ and the vector of attributes as $S$. The generative model for this dataset is the same as Example 3. The class-preserving generation and style transfer tasks were conducted in the same manner as MNIST. In addition, we also tried generating samples with manipulated attributes. Since the attribute encoder $f_3$ embeds $S$ in the Euclidean space, we could extrapolate input $S$ to decoder $g$ beyond 0 and 1. For this attribute manipulation task, we compared results with Fader Networks trained with a similar architecture. Fig. 4 shows sample images for all tasks. Although images of persons who were *not* in the training data were used, the WFAE could successfully generate images retaining the identity while employing other identity-invariant

---

[1]In this experiment there is no $Y$ and the digit class plays the role of $S$. So the generation is usual class-conditional one, hence the FID scores are lower for the WFAE as expected from visual comparison.

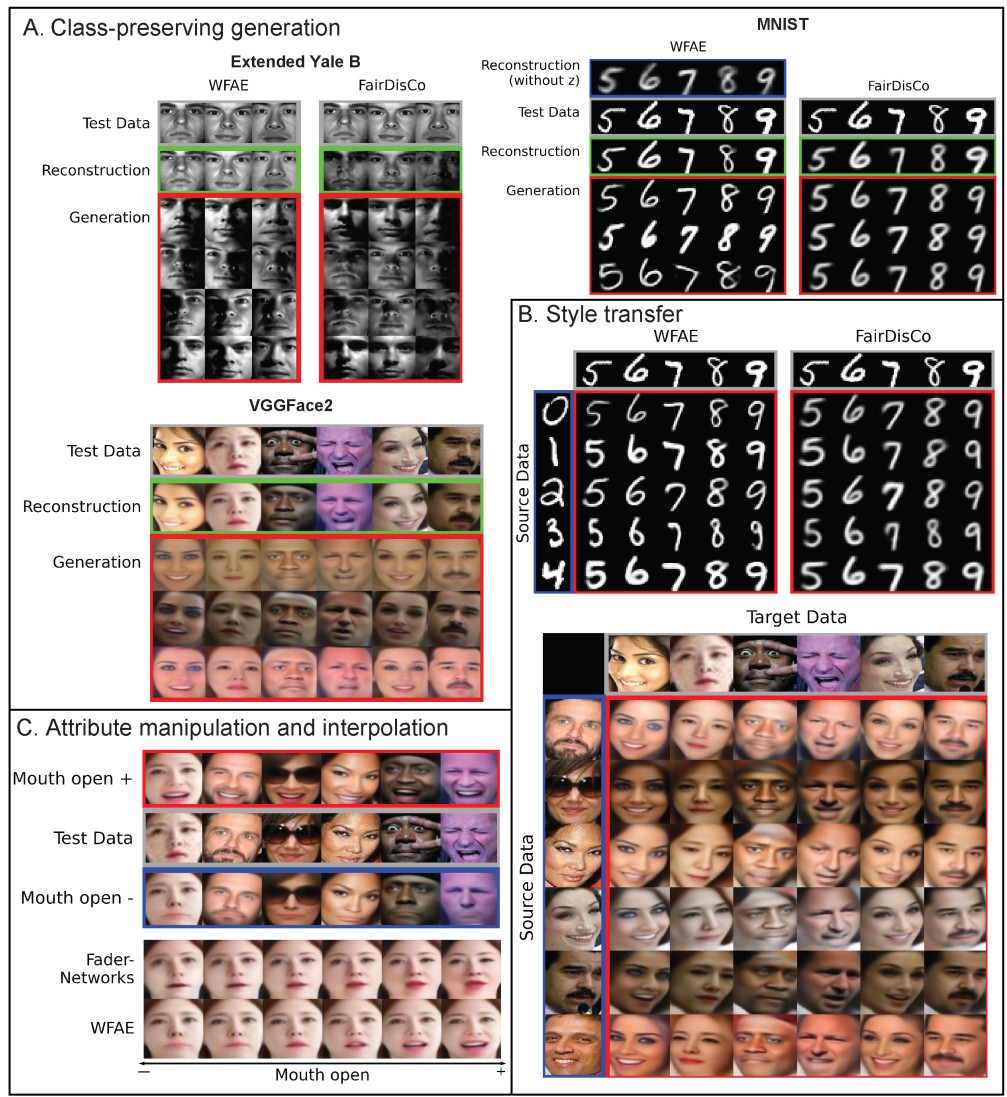

Figure 4: Conditional generation examples of WFAE

features, e.g., camera angle, lighting condition (panel A, bottom). Note in the style transfer task (panel B, bottom), the generated images possess the styles from the source data and tend to preserve the specified attribute of the target data. For example, the generated images tend to have open mouth if the target image has mouth wide open. In the attribute manipulation task, we could successfully generate images with the desired attributes changed. In panel C, letting the "Mouth Open" attribute positive produced decoded images having grinning mouth; making it negative produced images with lips all closed. For the Fader Networks, we extrapolated the attribute scores to a large magnitude as far as $\pm 400$, but it only caused deformation of the original image; see Remark 3.

# 6 Conclusion

We have shown that the WAE framework is rich enough to handle various conditional independence structures, leading to much more principled formulation of learning problems than the VAE counterparts. Importantly, a conditional independence structure imposes on the decoder automatically determines the encoder structure and the associated constraints. We hope this paper stimulates further research on extensions of WAEs in this direction, for instance, to complex hierarchical structures.

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

## A  PROOF OF THEOREM 1

**Proof 1** *Under the conditions of the theorem statement, the Monge-Kantorovich equivalence holds (see, e.g., Peyré & Cuturi, 2019, Theorem 2.1):*

$$W_2^2(P_X, P_G) = \inf_{T:\mathcal{X}\to\mathcal{X}:T_\sharp P_X = P_G} \mathbb{E}_{P_X} d^2(X, T(X)).$$

*Hence it suffices to show that*

$$\inf_{f:\mathcal{X}\to\mathcal{Z}:f_\sharp P_X=P_Z} \int_{\mathcal{X}} d^2(x,g(f(x)))dP_X = \inf_{T:\mathcal{X}\to\mathcal{X}:T_\sharp P_X=P_G} \int_{\mathcal{X}} d^p(x,T(x))dP_X$$

*or equivalently*

$$\{g \circ f : f : \mathcal{X} \to \mathcal{Z}, \ f_\sharp P_X = P_Z\} = \{T : \mathcal{X} \to \mathcal{X} : T_\sharp P_X = P_G\}.$$

*The forward inclusion $\subset$ holds since for any measurable $f : \mathcal{X} \to \mathcal{Z}$ such that $f_\sharp P_X = P_X f^{-1} = P_Z$ we have $g \circ f : \mathcal{X} \to \mathcal{X}$ measurable and for any Borel set $E \subset \mathcal{X}$*

$$(g \circ f)_\sharp P_X(E) = P_X(g \circ f)^{-1}(E) = P_X(f^{-1}(g^{-1}(E)))$$
$$= g_\sharp[P_X q^{-1}](E) = g_\sharp f_\sharp P_X(E) = g_\sharp P_Z(E) = P_G(E).$$

*For the backward inclusion $\supset$, suppose $T : \mathcal{X} \to \mathcal{X}$ is measurable and satisfies $T_\sharp P_X = P_G$. Since $g$ is injective, it has a left inverse $g^\dagger : \mathcal{X} \to \mathcal{Z}$. Let $f = g^\dagger \circ T$. Then for any Borel set $F \subset \mathcal{Z}$,*

$$f_\sharp P_X(F) = P_X(g^\dagger \circ T)^{-1}(F) = P_X(T^{-1}((g^\dagger)^{-1}(F)))$$
$$= T_\sharp P_X((g^\dagger)^{-1}(F))$$
$$= P_G((g^\dagger)^{-1}(F))$$
$$= g_\sharp P_Z((g^\dagger)^{-1}(F))$$
$$= P_Z(g^{-1}((g^\dagger)^{-1}(F)))$$
$$= P_Z((g^\dagger \circ g)^{-1}(F)) = P_Z(F),$$

*which completes the proof.*

## B   ADDITIONAL DETAILS FOR THE FAIR REPRESENTATION EXPERIMENT

Following Liu et al. (2022), fair representation experiment were held for the Adult Income and Health datasets, whose characteristics are described in Table 3. Note that all variables were categorized: one-hot encoding was used for variables with multiple category to make all data either 0 or 1. The encoder-decoder architecture of the network was adopted from (Louizos et al., 2016) (Table 4)

| Data | Training data size | Test Data Number | Covariate Dimension | Response Variable | Sensitive Variable |
|---|---|---|---|---|---|
| Adult Income | 30162 | 15060 | 115 | account | gender |
| Health | 44116 | 11030 | 254 | hospitalization | age |

Table 3: Information on categorical datasets for fair representation task.

# C FURTHER IMPLEMENTATION DETAILS

In the source code attached, settings for all experiments are gathered as configuration files in directory `configs/train_info`. All the network architecture are listed in `.py` files in `src/model` directory, and `model` and `architecture` keyword in the configuration file specifies which architecture to use among them. The running script `run.sh` that states which configuration was used for each experiment, managed by Hydra 1.1.1.Yadan (2019), can be found in the `experiments` directory. The prior $P_Z$ for the encoded $Z$ was set to be a normal distribution $N(0, 2I_l)$, where $l$ is the dimension of the latent space $\mathcal{Z}$. For the penalty divergences $\mathcal{D}_i$, we used the generative adversarial network (GAN) loss, which requires an additional discriminator (Tolstikhin et al., 2018). All the networks were trained using ADAM (Kingma & Ba, 2014) without any learning rate scheduling.

**Extended Yale B** The cropped version of the Extended Yale Face Database B dataset (Georghiades et al., 2001; Lee et al., 2005) were resized into a size of $128 \times 128$. The encoder-decoder architecture of the network had total of 18.5M parameters, and the discriminator architecture had 881 parameters (Table 5). After pre-training the $Y$-encoder with 2,100 iterations, we optimized the network for 5,200 iterations, which took about 40 minutes. The results were compared with the HSIC-contrained variational fair autoencoder (HCV, Table 6) and FairDisCo Table 7.

**MNIST** The encoder-decoder architecture of the network had 3.8M parameters, and the discriminator architecture had 7.4k parameters (Table 8). We pre-trained the $S$-encoder with 6,000 iterations, then optimized the rest of the network for 11,700 iterations, which took about half an hour. The results were compared with HCV and FairDisCo with the $S$ information available for decoding (Table 9 and Table 10).

**VGGFace2** The face region of the collected data were cropped and resized into a size of $128 \times 128$. The encoder-decoder architecture of the network had 88.4M parameters, and the discriminator architecture had 206k parameters (Table 11). We pre-trained the $(Y, S)$-encoder with 3,000 iterations, then optimized the rest of the network for 30,000 iterations, which took 16 hours. The results were compared with the Fader Network having an encoder-decoder architecture with 70.2M parameters and a discriminator architecture with 483k parameters (Table 12) trained for 20,000 iterations, which took 11 hours.

**Computing infrastructure** We trained the networks with Intel® Xeon® CPU E5-2650 v4 @ 2.20GHz processors and Nvidia Titan X Pascal GPUs with 12GB memory. For the VGGFace2 experiments, we trained the network using four GPUs; those for the other experiments were all trained using a single GPU. All the implementations were based on Python 3.6, PyTorch 1.10.2, PyTorch Lightning 1.5.10, and CUDA 10.2.

| Map | Layer | Operation | Filters | Batch norm | Activation | Linked layer |
|---|---|---|---|---|---|---|
| $f_2$ | 1 | Dense | 100 | Yes | ReLU | $X$ |
| | $\mu$ | Dense | $d_Z$ | No | - | $1, Y, S$ |
| | $\sigma^2$ | Dense | $d_Z$ | No | - | $1, Y, S$ |
| | Output $(Z_2\|X,Y,S)$ | Sample $Z_2\|X,Y,S$ | - | - | - | $\mu, \sigma^2$ |
| $g_1$ | 1 | Dense | 100 | Yes | ReLU | $g_2, S$ |
| | 2 | Dense | $d_X$ | No | Sigmoid | 1 |
| $g_2$ | 1 | Dense | 100 | Yes | ReLU | $Z_2, S$ |
| | $\mu$ | Dense | $d_Z$ | No | - | 1 |
| | $\sigma^2$ | Dense | $d_Z$ | No | - | 1 |
| | Output $(Z_1\|Z_2,Y)$ | Sample $Z_1\|Z_2,Y$ | - | - | - | $\mu, \sigma^2$ |
| Discriminator | 1 | Dense | $4d_Z$ | No | ReLU | $f_1$ |
| | 2 | Dense | $4d_Z$ | No | ReLU | 1 |
| | 3 | Dense | $4d_Z$ | No | ReLU | 2 |
| | 4 | Dense | $4d_Z$ | No | ReLU | 3 |
| | 5 | Dense | 1 | No | - | 4 |

Table 4: WFAE architecture for the fairness representation experiment: for the Adult income dataset, $d_X = 115, d_Z = 50$, and for the Health dataset, $d_X = 254, d_Z = 8$.

| Map | Layer | Operation | Filters | Kernel | Strides | Batch norm | Activation | Linked layer |
|---|---|---|---|---|---|---|---|---|
| $f_1$ | 1 | Convolution | 64 | 5x5 | 2x2 | Yes | ReLU | $X$ |
| | 2 | Convolution | 128 | 5x5 | 2x2 | Yes | ReLU | 1 |
| | 3 | Convolution | 256 | 5x5 | 2x2 | Yes | ReLU | 2 |
| | 4 | Convolution | 512 | 3x3 | 2x2 | Yes | ReLU | 3 |
| | 5 | Convolution | 1024 | 3x3 | 2x2 | Yes | ReLU | 4 |
| | 6 | Dense | 8 | - | - | - | - | 5 |
| $f_2$ | 1 | Convolution | 32 | 5x5 | 2x2 | Yes | ReLU | $X$ |
| | 2 | Convolution | 64 | 5x5 | 2x2 | Yes | ReLU | 1 |
| | 3 | Convolution | 128 | 5x5 | 2x2 | Yes | ReLU | 2 |
| | 4 | Convolution | 256 | 3x3 | 2x2 | Yes | ReLU | 3 |
| | 5 | Convolution | 512 | 3x3 | 2x2 | Yes | ReLU | 4 |
| | 6 | Dense | 2 | - | - | - | - | 5 |
| $g_1$ | 1 | Dense | 8x8x1024 | - | - | No | - | $g_2$ |
| | 2 | Transpose Convolution | 512 | 3x3 | 2x2 | Yes | LeakyReLU | 1 |
| | 3 | Transpose Convolution | 256 | 3x3 | 2x2 | Yes | LeakyReLU | 2 |
| | 4 | Convolution | 256 | 3x3 | 1x1 | Yes | LeakyReLU | 3 |
| | 5 | Transpose Convolution | 128 | 5x5 | 2x2 | Yes | LeakyReLU | 4 |
| | 6 | Transpose Convolution | 64 | 5x5 | 2x2 | Yes | LeakyReLU | 5 |
| | 7 | Convolution | 64 | 5x5 | 1x1 | Yes | LeakyReLU | 6 |
| | 8 | Convolution | 1 | 5x5 | 1x1 | No | Sigmoid | 7 |
| $g_2$ | 1 | Dense | 50 | - | - | Yes | LeakyReLU | $f_1, f_2$ |
| Discriminator | 1 | Dense | 16 | - | - | No | ReLU | $f_2$ |
| | 2 | Dense | 16 | - | - | No | ReLU | 1 |
| | 3 | Dense | 16 | - | - | No | ReLU | 2 |
| | 4 | Dense | 16 | - | - | No | ReLU | 3 |
| | 5 | Dense | 1 | - | - | No | - | 4 |

Table 5: WFAE architecture for the Extended Yale B dataset.

| Map | Layer | Operation | Filters | Kernel | Strides | Batch norm | Activation | Linked layer |
|---|---|---|---|---|---|---|---|---|
| $Q_{Z_1\|X,S}$ | 1 | Convolution | 64 | 5x5 | 2x2 | Yes | ReLU | $X$ |
| | 2 | Convolution | 128 | 5x5 | 2x2 | Yes | ReLU | 1 |
| | 3 | Convolution | 256 | 5x5 | 2x2 | Yes | ReLU | 2 |
| | 4 | Convolution | 512 | 3x3 | 2x2 | Yes | ReLU | 3 |
| | 5 | Convolution | 1024 | 3x3 | 2x2 | Yes | ReLU | 4 |
| | $\mu$ | Dense | 10 | - | - | No | - | 5, $S$ |
| | $\sigma^2$ | Dense | 10 | - | - | No | - | 5, $S$ |
| | Output $(Z_1\|X,S)$ | Sample $Z_1\|X,S$ | - | - | - | - | - | $\mu, \sigma^2$ |
| $Q_{Z2\|Z_1,Y}$ | 1 | Dense | 20 | - | - | Yes | ReLU | $Z_1, Y$ |
| | $\mu$ | Dense | 10 | - | - | No | - | 1 |
| | $\sigma^2$ | Dense | 10 | - | - | No | - | 1 |
| | Output $(Z_2\|Z_1,Y)$ | Sample $Z_2\|Z_1,Y$ | - | - | - | - | - | $\mu, \sigma^2$ |
| $Q_{Y\|Z_1}$ | 1 | Dense | 20 | - | - | Yes | ReLU | $Z_1$ |
| | 2 | Dense | 38 | - | - | No | - | 1 |
| $P_{Z_1\|Z_2,Y}$ | 1 | Dense | 20 | - | - | Yes | ReLU | $Z_2, Y$ |
| | $\mu$ | Dense | 10 | - | - | No | - | 1 |
| | $\sigma^2$ | Dense | 10 | - | - | No | - | 1 |
| | Output $(Z_1\|Z_2,Y)$ | Sample $Z_1\|Z_2,Y$ | - | - | - | - | - | $\mu, \sigma^2$ |
| $P_{X\|Z_1,S}$ | 1 | Dense | 8x8x1024 | - | - | No | - | $Z_1, S$ |
| | 2 | Transpose Convolution | 512 | 3x3 | 2x2 | Yes | ReLU | 1 |
| | 3 | Transpose Convolution | 256 | 3x3 | 2x2 | Yes | ReLU | 2 |
| | 4 | Convolution | 256 | 3x3 | 1x1 | Yes | ReLU | 3 |
| | 5 | Transpose Convolution | 128 | 5x5 | 2x2 | Yes | ReLU | 4 |
| | 6 | Transpose Convolution | 64 | 5x5 | 2x2 | Yes | ReLU | 5 |
| | 7 | Convolution | 64 | 5x5 | 1x1 | Yes | ReLU | 6 |
| | 8 | Convolution | 1 | 5x5 | 1x1 | No | Sigmoid | 7 |

Table 6: HCV architecture for the Extended Yale B dataset.

| Map | Layer | Operation | Filters | Kernel | Strides | Batch norm | Activation | Linked layer |
|---|---|---|---|---|---|---|---|---|
| $Q_{Z_2\|X,S}$ | 1 | Convolution | 32 | 5x5 | 2x2 | Yes | ReLU | $X$ |
| | 2 | Convolution | 64 | 5x5 | 2x2 | Yes | ReLU | 1 |
| | 3 | Convolution | 128 | 5x5 | 2x2 | Yes | ReLU | 2 |
| | 4 | Convolution | 256 | 3x3 | 2x2 | Yes | ReLU | 3 |
| | 5 | Convolution | 512 | 3x3 | 2x2 | Yes | ReLU | 4 |
| | $\mu$ | Dense | 2 | - | - | No | - | 5, $S$ |
| | $\sigma^2$ | Dense | 2 | - | - | No | - | 5, $S$ |
| | Output $(Z_2\|X,S)$ | Sample $Z_2\|X,S$ | - | - | - | - | - | $\mu, \sigma^2$ |
| $P_{Z_1\|Z_2,Y}$ | 1 | Dense | 50 | - | - | Yes | ReLU | $Z_2, Y$ |
| | $\mu$ | Dense | 10 | - | - | No | - | 1 |
| | $\sigma^2$ | Dense | 10 | - | - | No | - | 1 |
| | Output $(Z_1\|Z_2,Y)$ | Sample $Z_1\|Z_2,Y$ | - | - | - | - | - | $\mu, \sigma^2$ |
| $P_{X\|Z_1,S}$ | 1 | Dense | 8x8x1024 | - | - | No | - | $Z_1, S$ |
| | 2 | Transpose Convolution | 512 | 3x3 | 2x2 | Yes | ReLU | 1 |
| | 3 | Transpose Convolution | 256 | 3x3 | 2x2 | Yes | ReLU | 2 |
| | 4 | Convolution | 256 | 3x3 | 1x1 | Yes | ReLU | 3 |
| | 5 | Transpose Convolution | 128 | 5x5 | 2x2 | Yes | ReLU | 4 |
| | 6 | Transpose Convolution | 64 | 5x5 | 2x2 | Yes | ReLU | 5 |
| | 7 | Convolution | 64 | 5x5 | 1x1 | Yes | ReLU | 6 |
| | 8 | Convolution | 1 | 5x5 | 1x1 | No | Sigmoid | 7 |

Table 7: FairDisCo architecture for the Extended Yale B dataset.

| Map | Layer | Operation | Filters | Kernel | Strides | Batch norm | Activation | Linked layer |
|---|---|---|---|---|---|---|---|---|
| $f_2$ | 1 | Convolution | 64 | 4x4 | 2x2 | Yes | ReLU | $X$ |
| | 2 | Convolution | 128 | 4x4 | 2x2 | Yes | ReLU | 1 |
| | 3 | Convolution | 256 | 4x4 | 2x2 | Yes | ReLU | 2 |
| | 4 | Convolution | 512 | 4x4 | 2x2 | Yes | ReLU | 3 |
| | 5 | Dense | 6 | - | - | No | - | 4 |
| $f_3$ | 1 | Convolution | 32 | 4x4 | 2x2 | Yes | ReLU | $X$ |
| | 2 | Convolution | 32 | 4x4 | 1x1 | Yes | ReLU | 1 |
| | 3 | Convolution | 64 | 4x4 | 2x2 | Yes | ReLU | 2 |
| | 4 | Convolution | 64 | 4x4 | 1x1 | Yes | ReLU | 3 |
| | 5 | Dense | 6 | - | - | - | - | 4 |
| $g$ | 1 | Dense | 7x7x256 | - | - | No | - | $f_2, f_3$ |
| | 2 | Transpose Convolution | 128 | 4x4 | 2x2 | Yes | LeakyReLU | 1 |
| | 3 | Transpose Convolution | 64 | 4x4 | 2x2 | Yes | LeakyReLU | 2 |
| | 4 | Convolution | 64 | 4x4 | 1x1 | Yes | LeakyReLU | 3 |
| | 5 | Convolution | 1 | 4x4 | 1x1 | No | Sigmoid | 4 |
| Discriminator | 1 | Dense | 48 | - | - | No | ReLU | $f_3$ |
| | 2 | Dense | 48 | - | - | No | ReLU | 1 |
| | 3 | Dense | 48 | - | - | No | ReLU | 2 |
| | 4 | Dense | 48 | - | - | No | ReLU | 3 |
| | 5 | Dense | 1 | - | - | No | - | 4 |

Table 8: WFAE architecture for the MNIST dataset.

| Map | Layer | Operation | Filters | Kernel | Strides | Batch norm | Activation | Linked layer |
|---|---|---|---|---|---|---|---|---|
| $Q_{Z\|X,S}$ | 1 | Convolution | 64 | 4x4 | 2x2 | Yes | ReLU | $X$ |
| | 2 | Convolution | 128 | 4x4 | 2x2 | Yes | ReLU | 1 |
| | 3 | Convolution | 256 | 4x4 | 2x2 | Yes | ReLU | 2 |
| | 4 | Convolution | 512 | 4x4 | 2x2 | Yes | ReLU | 3 |
| | 5 | Dense | 100 | - | - | Yes | ReLU | 4, $S$ |
| | $\mu$ | Dense | 6 | - | - | No | - | 5 |
| | $\sigma^2$ | Dense | 6 | - | - | No | - | 5 |
| | Output $(Z_1\|X, S)$ | Sample $Z_1\|X, S$ | - | - | - | - | - | $\mu, \sigma^2$ |
| $P_{X\|Z,S}$ | 1 | Dense | 7x7x256 | - | - | No | - | $Z, S$ |
| | 2 | Transpose Convolution | 128 | 4x4 | 2x2 | Yes | ReLU | 1 |
| | 3 | Transpose Convolution | 64 | 4x4 | 2x2 | Yes | ReLU | 2 |
| | 4 | Convolution | 64 | 4x4 | 1x1 | Yes | ReLU | 3 |
| | 5 | Convolution | 1 | 4x4 | 1x1 | No | Sigmoid | 4 |

Table 9: HCV architecture for the MNIST dataset.

| Map | Layer | Operation | Filters | Kernel | Strides | Batch norm | Activation | Linked layer |
|---|---|---|---|---|---|---|---|---|
| $Q_{Z\|X,S}$ | 1 | Convolution | 64 | 4x4 | 2x2 | Yes | ReLU | $X$ |
| | 2 | Convolution | 128 | 4x4 | 2x2 | Yes | ReLU | 1 |
| | 3 | Convolution | 256 | 4x4 | 2x2 | Yes | ReLU | 2 |
| | 4 | Convolution | 512 | 4x4 | 2x2 | Yes | ReLU | 3 |
| | $\mu$ | Dense | 6 | - | - | No | - | 4 |
| | $\sigma^2$ | Dense | 6 | - | - | No | - | 4 |
| | Output $(Z_1\|X, S)$ | Sample $Z_1\|X, S$ | - | - | - | - | - | $\mu, \sigma^2$ |
| $P_{X\|Z,S}$ | 1 | Dense | 7x7x256 | - | - | No | - | $Z, S$ |
| | 2 | Transpose Convolution | 128 | 4x4 | 2x2 | Yes | ReLU | 1 |
| | 3 | Transpose Convolution | 64 | 4x4 | 2x2 | Yes | ReLU | 2 |
| | 4 | Convolution | 64 | 4x4 | 1x1 | Yes | ReLU | 3 |
| | 5 | Convolution | 1 | 4x4 | 1x1 | No | Sigmoid | 4 |

Table 10: FairDisCo architecture for the MNIST dataset.

| Map | Layer | Operation | Filters | Kernel | Strides | Batch norm | Activation | Linked layer |
|---|---|---|---|---|---|---|---|---|
| $(f_1, f_2)$ | 1 | Convolution | 128 | 5x5 | 2x2 | Yes | ReLU | $X$ |
| | 2 | Convolution | 256 | 5x5 | 2x2 | Yes | ReLU | 1 |
| | 3 | Convolution | 512 | 5x5 | 2x2 | Yes | ReLU | 2 |
| | 4 | Convolution | 1024 | 5x5 | 2x2 | Yes | ReLU | 3 |
| | 5 | Dense | 71 | - | - | - | - | 4 |
| $f_3$ | 1 | Convolution | 128 | 5x5 | 2x2 | Yes | ReLU | $X$ |
| | 2 | Convolution | 128 | 5x5 | 1x1 | Yes | ReLU | 1 |
| | 3 | Convolution | 256 | 5x5 | 2x2 | Yes | ReLU | 2 |
| | 4 | Convolution | 256 | 5x5 | 1x1 | Yes | ReLU | 3 |
| | 5 | Convolution | 512 | 5x5 | 2x2 | Yes | ReLU | 4 |
| | 6 | Convolution | 512 | 3x3 | 1x1 | Yes | ReLU | 5 |
| | 7 | Convolution | 1024 | 3x3 | 2x2 | Yes | ReLU | 6 |
| | 8 | Convolution | 1024 | 3x3 | 1x1 | Yes | ReLU | 7 |
| | 9 | Dense | 32 | - | - | - | - | 8 |
| $g$ | 1 | Dense | 8x8x1024 | - | - | No | - | $f_1, f_2, f_3$ |
| | 2 | Transpose Convolution | 512 | 5x5 | 2x2 | Yes | LeakyReLU | 1 |
| | 3 | Residual Block | 512 | 5x5, 1x1 | 1x1 | Yes | LeakyReLU | 2 |
| | 4 | Transpose Convolution | 256 | 5x5 | 2x2 | Yes | LeakyReLU | 3 |
| | 5 | Residual Block | 256 | 5x5, 1x1 | 1x1 | Yes | LeakyReLU | 4 |
| | 6 | Transpose Convolution | 128 | 5x5 | 2x2 | Yes | LeakyReLU | 5 |
| | 7 | Residual Block | 128 | 3x3, 1x1 | 1x1 | Yes | LeakyReLU | 6 |
| | 8 | Transpose Convolution | 64 | 5x5 | 2x2 | Yes | LeakyReLU | 7 |
| | 9 | Residual Block | 64 | 3x3, 1x1 | 1x1 | Yes | LeakyReLU | 8 |
| | 10 | Convolution | 3 | 3x3 | 1x1 | No | Sigmoid | 9 |
| Discriminator | 1 | Dense | 256 | - | - | No | ReLU | $f_3$ |
| | 2 | Dense | 256 | - | - | No | ReLU | 1 |
| | 3 | Dense | 256 | - | - | No | ReLU | 2 |
| | 4 | Dense | 256 | - | - | No | ReLU | 3 |
| | 5 | Dense | 1 | - | - | No | - | 4 |

Table 11: WFAE architecture for the VGGFace2 dataset.

| Map | Layer | Operation | Filters | Kernel | Strides | Batch norm | Activation | Linked layer |
|---|---|---|---|---|---|---|---|---|
| Encoder $f$ | 1 | Convolution | 128 | 5x5 | 2x2 | Yes | ReLU | $X$ |
| | 2 | Convolution | 128 | 5x5 | 1x1 | Yes | ReLU | 1 |
| | 3 | Convolution | 256 | 5x5 | 2x2 | Yes | ReLU | 2 |
| | 4 | Convolution | 256 | 5x5 | 1x1 | Yes | ReLU | 3 |
| | 5 | Convolution | 512 | 5x5 | 2x2 | Yes | ReLU | 4 |
| | 6 | Convolution | 512 | 3x3 | 1x1 | Yes | ReLU | 5 |
| | 7 | Convolution | 1024 | 3x3 | 2x2 | Yes | ReLU | 6 |
| | 8 | Convolution | 1024 | 3x3 | 1x1 | Yes | ReLU | 7 |
| | 9 | Dense | 96 | - | - | - | - | 8 |
| Decoder $g$ | 1 | Dense | 8x8x1024 | - | - | No | - | $S, Z$ |
| | 2 | Transpose Convolution | 512 | 5x5 | 2x2 | Yes | LeakyReLU | 1 |
| | 3 | Residual Block | 512 | 5x5, 1x1 | 1x1 | Yes | LeakyReLU | 2 |
| | 4 | Transpose Convolution | 256 | 5x5 | 2x2 | Yes | LeakyReLU | 3 |
| | 5 | Residual Block | 256 | 5x5, 1x1 | 1x1 | Yes | LeakyReLU | 4 |
| | 6 | Transpose Convolution | 128 | 5x5 | 2x2 | Yes | LeakyReLU | 5 |
| | 7 | Residual Block | 128 | 3x3, 1x1 | 1x1 | Yes | LeakyReLU | 6 |
| | 8 | Transpose Convolution | 64 | 5x5 | 2x2 | Yes | LeakyReLU | 7 |
| | 9 | Residual Block | 64 | 3x3, 1x1 | 1x1 | Yes | LeakyReLU | 8 |
| | 10 | Convolution | 3 | 3x3 | 1x1 | No | Sigmoid | 9 |
| Discriminator | 1 | Dense | 384 | - | - | No | ReLU | $Z$ |
| | 2 | Dense | 384 | - | - | No | ReLU | 1 |
| | 3 | Dense | 384 | - | - | No | ReLU | 2 |
| | 4 | Dense | 384 | - | - | No | ReLU | 3 |
| | 5 | Dense | 7 | - | - | No | - | 4 |

Table 12: Fader Network architecture for the VGGFace2 dataset.

