# OpenReview forum: "Wasserstein Fair Autoencoders"
_ICLR.cc/2023/Conference — Submitted to ICLR 2023_

### Official Review · Reviewer_Zcji · 2022-10-22

**Confidence:** 4
**Correctness:** 2
**Technical Novelty And Significance:** 2
**Empirical Novelty And Significance:** 2
**Recommendation:** 3

**Clarity, Quality, Novelty And Reproducibility:**

* Clarity: The paper is not easy to follow since it uses a lot of notations.
* Quality: The paper does not compare the proposed framework with recent frameworks.
* Novelty: The paper uses directly the techniques from the Wasserstein autoencoder ( WAE) paper.
* Reproducibility: The code is submitted, and neural network architectures are reported carefully. However, some settings such as the choice of ground metric and divergences seem to be missing.

**Strength And Weaknesses:**

## Strength

* The paper is the first work that extends WAE to the setting of the "M1" and "M2" models.
* The experimental results indicate that the proposed framework is better than the previous approach such as variational fair AE and FairDisCo.

## Weaknesses

* The writing is hard to follow. The author can simply replace complicated notations by showing the graphical model of the inference/recognition/encoding model.
* The chosen ground metric e.g., $\tilde{d}$ in Example 1 is not a valid metric on the joint space of supports of random variables for generalized choices of  $d$ and $d'$.  Therefore. the final objective is not the "Wasserstein" distance. The specific settings of those distances are missing in experiments.
* The duality gap could be large for the Lagrange duality. The tuning of various Lagrange multipliers is a negative point.  Moreover, the choices of the divergences e.g., $D$ and $H$ in Example 1 are not discussed.
* The baselines of experiments are not strong. There are more recent works on the same problem e.g., [1],[2]. Also, the performance of the proposed framework is not clearly superior compared to the chosen baselines e.g., Figure 2, Table. 2.

[1] Hierarchical VampPrior Variational Fair Auto-Encoder, Philip Botros and Jakub M. Tomczak
[2] Flexibly Fair Representation Learning by Disentanglement, Elliot Creager et al.

**Summary Of The Paper:**

The paper extends the Wasserstein autoencoder framework for the unconditional generative model to the setting of having structural constraints e.g., conditional independence on the latent variables (representation, or factors). In greater detail, the authors consider three examples of having additional information to the generator. The first example is the "M2" model (Kingma et al., 2014). In this setting, there exists an observed nuisance variable $S$, and the latent variable $Z$ wants to encode the representation invariant to the unwanted variation in $S$. The second example is the extension of the variational fair autoencoder (an extension of the “M1 + M2” semi-supervised model) that has partially observed label variable $Y$ in addition to the nuisance variable  $S$. The final example is the extension of the first example with two independent nuisance variables that can be missing. The main modeling techniques in the paper are factorizing the couplings between the joint observed data variables distribution and the joint model variables distribution by the chain rule and matching the aggregated posterior to the prior (which is used in WAE and is equivalent to Kantorovich optimal transport via the gluing lemma). The ground metric on the joint space is the square root of the square of distances on marginal variables. The authors compare the proposed framework with the HSICconstrained VFAE (HCV) and the FairDisCo in learning fair representation on Adult Income and Health datasets, learning invariant representation on the Extended Yale B dataset, and conditional generation on MNIST and VGGFace2 datasets.

**Summary Of The Review:**

The paper should include a graphical model for the inference model (encoder) and compare the proposed framework with recent papers (mentioned in the weaknesses part).

---

### Official Review · Reviewer_ernN · 2022-11-01

**Confidence:** 2
**Correctness:** 3
**Technical Novelty And Significance:** 2
**Empirical Novelty And Significance:** 2
**Recommendation:** 5

**Clarity, Quality, Novelty And Reproducibility:**

The paper is easy to read and the quality is high. The novelty is not that high, but the reviewer find the findings of importance and interest to the research community.

**Strength And Weaknesses:**

Thank you for a very interesting read. The research is of high interest to the research community and shows a clear strength of using WAEs. To my knowledge and review, the theory is sound. The experiments are meaningful and provide good empirical evidence for the claims in the paper.

The weakness from the initial review regards the strong advocation that this is in contrast to VAEs. I agree with the fact that many research papers have employed tricks on the objective function that are far from principled in the VAEs.

Regarding the comparison to VAEs:
1) Have you, in fact, shown that WFAEs are preferred over the VAE framework? If so, a response in the rebuttal would be much appreciated.
2) Is the comparison to the VAE papers fair? For example, some of the very deep VAE papers, e.g., "Very Deep VAEs ..." by Child et al. or "BIVA ..." by Maaloe et al. shows principled approaches to the VAE in cohesion to image generation examples that could suggest that they learn structural constraints on the latent variables. What would happen if you employ a structural constraint on these models only by altering the graphical model (no changes on the objective function) and how would that compare to the WFAE?

Regarding the general empirical findings:
- It would be useful to see many more examples of conditional generation examples, e.g., figure 5 in "Autoencoding beyond pixels using a learned similarity metric" by Larsen et al.
- What about semi-supervised classification evaluations? Wouldn't it be possible to report on these and compare to state-of-the-art?

**Summary Of The Paper:**

The authors argue that there is a principled way to handle structural constraints on the latent variables in Wasserstein AutoEncoders. They argue that this is in contrast to the VAE where numerous tricks have been employed on the objective function in order to impose similar structural constraints, e.g., the semi-supervised work and beta-VAE.

**Summary Of The Review:**

Interesting paper. Concerns can be alleviated through a rebuttal of the comparison to VAEs and by expanding on the empirical finding.

---

### Official Review · Reviewer_4HcR · 2022-11-04

**Confidence:** 4
**Correctness:** 1
**Technical Novelty And Significance:** 1
**Empirical Novelty And Significance:** 1
**Recommendation:** 1

**Clarity, Quality, Novelty And Reproducibility:**

Clarity
The writing in general is clear and the analysis is easy to follow. All terms, notations, and steps are well described.

The main thesis of the paper is hard to follow, especially throughout the main body of the paper in the examples.

Quality
The quality of the paper is not up to the typical standard at ICLR. The main claim is not supported by the text, and it's not clear what else is being provided.

Novelty
The authors correct a previous theorem in prior work, but otherwise it is not clear what the novel observation and value proposition is.

Reproducibility
Code is provided and with the appendix, seems to be sufficient to replicate the experiments but I have not gone through in detail.

**Strength And Weaknesses:**

Strengths
1) The paper is detailed in its derivation for the numerous examples it presents.

2) The paper does a good job of setting up the problem and describing necessary background in a way that flows nicely with the story of the paper.

3) The notation, analysis, and arguments are typically very clear, and a careful reader would be able to follow most of the discussion with mild background in Wasserstein method and VAEs.

Weaknesses
1) The paper does not provide a solution for the problem and claim suggested in the abstract or introduction. The authors claim that penalties and constraints can be derived directly from, or induced from, the conditional independence structure of the WAE. The introduction describes a number of prior literature that use a variety of metrics to enforce or push independence, and claim that none of these are a ``principled way ofo imposing the encoder structure". The end of the preliminaries describes a funtion $\delta$ that already has a new, arbitrary divergence $\mathcal{D}$. The core of the paper, the three examples presented, all have additional regularization/constraint terms arbitrarily appended to their objectives, explicitly suggesting they can take the form of an MMD loss, GAN loss, HSIC loss, etc.

2) There is no general algorithmic or analytic procedure presented that might generalize past the three problems described here. The experimental setups shown follow the examples shown, but it's not clear what the examples are providing over the existing methods when the arbitrary regularizers are not added.

3) The experimental evaluation in general is not very convincing. The most important piece in experimental setups similar to this is choosing the regularization weights. This can drastically change the results of all models tested, and no discussion is provided that clarifies how or which weights were chosen.

**Summary Of The Paper:**

The paper proposes a new analysis of Wasserstein Autoencoders and claims the new analysis reveals a learning objective form that naturally can be optimized without the addition of ``ad-hoc" penalties. The paper describes this anaylsis and presents a full derivation of their construction in three prototypical generative model examples with side or nuisance information. They evaluate their method in a few settings where conditional generative models are deployed.

**Summary Of The Review:**

It is not clear what the paper is proposing and the main claim is not justified at all. The paper seems to have derived some formulations and then tacked on arbitrary regularizers, completely antithetical to the claims in the abstract and introduction.

---

### Decision · Program_Chairs · 2023-01-20

**Decision:**

Reject

**Justification For Why Not Higher Score:**

All reviewers agree that this paper is a clear reject and the authors do not provide responses to reviewers' questions.

**Justification For Why Not Lower Score:**

N/A

**Metareview: Summary, Strengths And Weaknesses:**

All reviewers agree that this paper is a clear reject and the authors do not provide responses to reviewers' questions.